# Cancer-associated TERT promoter mutations abrogate telomerase silencing

Kunitoshi Chiba, Joshua Z Johnson, Jacob M Vogan, Tina Wagner, John M Boyle, Dirk Hockemeyer*

Department of Molecular and Cell Biology, University of California, Berkeley, Berkeley, United States

**Abstract** Mutations in the human telomerase reverse transcriptase (TERT) promoter are the most frequent non-coding mutations in cancer, but their molecular mechanism in tumorigenesis has not been established. We used genome editing of human pluripotent stem cells with physiological telomerase expression to elucidate the mechanism by which these mutations contribute to human disease. Surprisingly, telomerase-expressing embryonic stem cells engineered to carry any of the three most frequent TERT promoter mutations showed only a modest increase in TERT transcription with no impact on telomerase activity. However, upon differentiation into somatic cells, which normally silence telomerase, cells with TERT promoter mutations failed to silence TERT expression, resulting in increased telomerase activity and aberrantly long telomeres. Thus, TERT promoter mutations are sufficient to overcome the proliferative barrier imposed by telomere shortening without additional tumor-selected mutations. These data establish that TERT promoter mutations can promote immortalization and tumorigenesis of incipient cancer cells.

## Introduction

Activation of telomerase is the critical step for the immortalization of more than 90% of all human tumors (*Greider and Blackburn, 1985*; *Counter, 1992*; *Kim et al., 1994*). Non-coding mutations in the promoter of the catalytic subunit of telomerase (TERT) emerged recently as one of the most prevalent mutations in human cancer (*Bojesen et al., 2013*; *Horn et al., 2013*; *Huang et al., 2013*; *Killela et al., 2013*; *Fredriksson et al.,2014*; *Weinhold et al., 2014*). Interestingly, all TERT promoter mutations associated with cancer formation thus far generate novel binding sites for the ETS (E26 transformation-specific) family of transcription factors and are located close to the translational start site of TERT (e.g., −57A/C, −124C/T, and −146C/T) (*Horn et al., 2013*; *Huang et al., 2013*). Transient transfection experiments using ectopic TERT Luciferase-reporter constructs suggest that TERT promoter mutations can increase TERT transcription by 1.5–2 fold when assayed in tumor cells (*Horn et al., 2013*; *Huang et al., 2013*). To date, the physiological events that select for these specific mutations are still unclear, as they have been mostly investigated for their impact in tumor cell lines that are already immortal, maintain telomere length, and have aberrant karyotypes. These tumor cell lines have sufficient telomerase activity to maintain an immortal phenotype, but so do tumor cells without these TERT promoter mutations. Thus, changes in telomerase levels and telomere length provide incomplete information regarding the functional differences between cells that do or do not carry TERT promoter mutations.

In untransformed human tissues, telomerase activity is restricted to embryonic cells and some adult stem cell or progenitor compartments due to transcriptional silencing of TERT upon differentiation (*Gunes and Rudolph, 2013*; *Aubert, 2014*). As a consequence, differentiated somatic cells undergo progressive telomere shortening with cell division, which limits their proliferative capacity and has thus been proposed as a tumor suppressor mechanism

*For correspondence:
hockemeyer@berkeley.edu

Competing interests: The authors declare that no competing interests exist.

**eLife digest** The bulk of the DNA in the human genome is divided between 23 pairs of chromosomes. The ends of these chromosomes contain a repetitive stretch of DNA known as a telomere. Every time a cell divides, a portion of the telomere is lost and can be restored by an enzyme called telomerase.

If the telomeres shorten below a critical length, the cell can no longer divide and eventually dies. Thus, long telomeres increase the number of times a cell can divide. In the majority of human cells—with the exception of stem cells—telomerase activity is absent due to the down regulation of the active protein component (called TERT) after birth. Therefore, the telomeres in these cells shorten after each cell division. However, 90% of human cancers have very high TERT activity, which enables them to divide continuously to drive tumor growth.

Genes are sections of DNA that code for proteins and other molecules. The start of a gene contains a region known as the promoter, which controls when and where in the body the gene is active. Cancer cells often contain mutations in the promoter of the gene that encodes TERT. However, it remains poorly understood how these mutations lead to the formation of tumors.

Chiba et al. have now used a technique called genome editing to introduce mutations that are commonly found in cancer cells into the promoter of the gene for TERT in human embryonic stem cells. Unexpectedly, these changes did not increase the activity of the telomerase enzyme in these cells, nor did they increase the length of the telomeres.

Chiba et al. next caused these genetically engineered stem cells to develop into more specialized cell types—such as nerve cells. These 'differentiated' cells normally silence the gene that encodes TERT, but the mutations prevented the gene from being silenced. This led to abnormally high levels of telomerase activity and long telomeres. The experiments also showed that TERT activity in these cells was similar to that found in cancer cells that can divide indefinitely.

Cells containing the promoter mutations were then injected into mice. The cells formed a mass of tumors that contained very long telomeres. These results together suggest that cancer-causing mutations in the gene for TERT stop this gene from being properly silenced in more specialized cells, and that this, on its own, can promote the formation of tumors. These findings are likely to underpin future efforts to treat cancers by targeting the expression and activity of the telomerase enzyme.

(*Wright et al., 1996*). Critically short telomeres are detected as sites of DNA damage leading to cell death or replicative senescence (*Palm and de Lange, 2008*). Long-term inhibition of TERT (*Herbert et al., 1999*) or interference with telomerase recruitment to telomeres (*Nakashima et al., 2013*; *Sexton et al., 2014*) lead to cell death in telomerase-positive cancer and stem cells. Inversely, ectopic telomerase expression is sufficient to immortalize normal human fibroblasts by allowing them to bypass senescence (*Bodnar et al., 1998*; *Morales et al., 1999*). Since the discovery of telomerase reactivation in cancer, many *cis*-regulatory elements and corresponding transcription factors have been suggested to contribute to the regulation of TERT in healthy cells and its aberrant expression in tumor cells (*Greenberg et al., 1999*; *Ducrest et al., 2002*; *Lin and Elledge, 2003*; *Kyo et al., 2008*). GWAS analysis identified a specific set of TERT promoter mutations in melanomas that all occur in a very small region close to the transcriptional start site and each results in novel putative TTCCGG- ETS binding sites (*Horn et al., 2013*; *Huang et al., 2013*). While ETS-factors are a large family of transcription factors that can recognize this binding site, recent data suggest that TERT promoter mutations are bound predominantly by GABP (*Bell et al., 2015*). This specificity does not appear restricted to melanomas as the same TERT promoter mutations have emerged as a major driver in a multitude of human solid tumors (*Heidenreich et al., 2014*), including glioblastomas, medulloblastomas, carcinomas of the bladder, urothelial cancer (*Borah et al., 2015*), thyroid and squamous cell carcinomas of the tongue, as well as in liposarcomas and hepatocellular carcinomas (*Heidenreich et al., 2014*). Based on this tumor spectrum, TERT promoter mutations have been hypothesized to preferentially promote tumor progression in tissues with relatively low rates of self-renewal (*Killela et al., 2013*). Several studies have suggested that TERT promoter mutations can provide a biomarker to stratify human cancer subtypes (*Heidenreich et al., 2014*; *Borah et al., 2015*). However, the mechanism by which these

mutations promote tumor formation is unknown. The key outstanding questions are: (1) whether TERT promoter mutations are sufficient to immortalize cells and (2) why TERT promoter mutations occur in specific tumors subtypes.

Here we address these questions by genetically engineering human embryonic stem cells (hESCs) to carry the three most prevalent cancer-associated TERT promoter mutations in an isogenic background. The impact of these mutations was studied by measuring their effect on TERT expression, telomerase activity, and telomere length in stem cells as well as in differentiated cell types. We demonstrate that two out of three cancer-associated TERT mutations caused no effect and only the most prevalent promoter mutations mildly increased TERT levels in hESCs, which did not result in significantly increased telomerase activity. We find that increased TERT expression is not functionally linked to an increase in active telomerase, as TR, the telomerase RNA component, but not TERT is limiting in hESCs. However, the importance of these mutations in tumorigenesis becomes clear when hESCs are differentiated into normally telomerase-negative cells. Under these conditions all cancer-associated TERT mutations prevent repression of TERT, resulting in a retention of telomerase activity relative to wild-type differentiated cells. Ultimately, the resulting TERT expression led to aberrant telomerase enzymatic activity in terminally differentiated cells and abnormally long telomeres, thereby bypassing the telomere shortening tumor suppressor pathway.

## Results

### Analyzing cancer-associated TERT promoter mutations in hESCs

We aimed to understand the molecular basis by which the cancer-associated TERT mutations impact telomerase biology. To address this question, we employed CAS9- (clustered regularly interspaced short palindromic repeats (CRISPR)/CRISPR-associated systems 9) (*Jinek et al., 2012*) mediated genome editing to derive human pluripotent stem cells (WIBR#3) that carry TERT promoter mutations at the endogenous TERT locus. Initially we attempted conventional donor-based genome editing strategies with sgRNAs targeting sequences proximal to the targeting site. These attempts were however unsuccessful, likely due to the TERT promoter mutations being in a genomic region with ~80% GC content. This non-random base composition does not allow for the design of specific sgRNAs without a large number of potential off-targets. We tested several sgRNAs in proximity to TERT promoter mutations and found them to be toxic to cells shortly after transfection into cancer cells and human primary fibroblasts. We overcame this challenge by employing a two-step targeting approach (*Figure 1A*). In a first editing step we homozygously deleted a 1.5 kb region in the TERT gene using two sgRNAs that cut at positions −1462 and +67 relative to the first ATG (*Figure 1A*). In a second editing step we reintroduced the deleted region either with or without the promoter mutations into the endogenous TERT locus (*Figure 1A,B*).

As the first deletion step removed the translational start site as well as some coding sequence of TERT, this targeting strategy resulted in telomerase-negative hESCs (TERT$^{\Delta/\Delta}$) (*Figure 1C*). Correct targeting was confirmed by Southern blot and PCR-sequencing of the genomic deletion (*Figure 1—figure supplement 1A,B*). As expected from our previous characterization of TERT$^{-/-}$ hESCs (*Sexton et al., 2014*), these cells proliferated normally for 3 to 4 months, followed by cell death due to progressive telomere shortening with no survivors after 140 days (*Sexton et al., 2014*). In a second targeting step we edited the newly formed genomic site with a specific sgRNA spanning the new junction (−1462 to +67) to reinsert and restore the deleted region with either the wild-type promoter or an altered region containing the most frequent cancer-associated TERT promoter mutations: −57A/C, −124C/T, or −146C/T (*Figure 1B*). This complementation approach restored the TERT gene and cellular viability of targeted cells. Accordingly, cells with a restored TERT gene gradually outcompeted none-rescued parental TERT$^{\Delta/\Delta}$ cells and lead to substantial telomere elongation by the time untargeted TERT$^{\Delta/\Delta}$ hESCs had died (*Figure 1D*). This complementation strategy therefore successfully generated hESCs that differed exclusively at the TERT locus by expressing TERT either from its wild-type promoter or from a promoter that contained one of the cancer-associated point mutations.

We first analyzed the impact of the TERT promoter mutations on TERT mRNA levels by qRT-PCR until cultures established stable TERT expression levels and all TERT$^{\Delta/\Delta}$ had died (*Figure 1E*). This analysis revealed that the most frequent TERT mutation (−124C/T) resulted in a 2–3-fold increase in

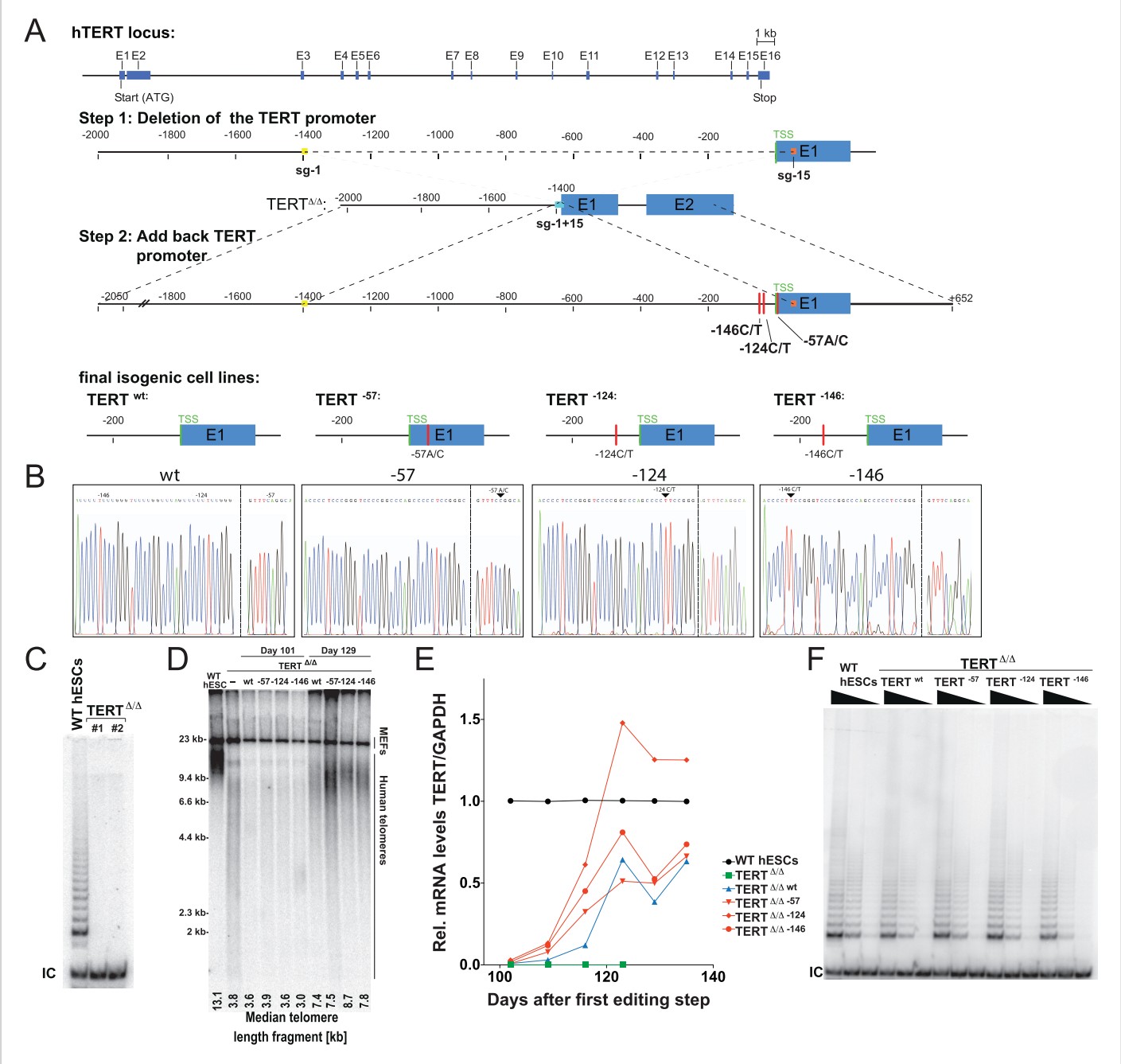

**Figure 1**. Generation of isogenic TERT promoter mutation-containing hESCs reveals a modest increase of TERT expression only for the −124C/T mutation. (**A**) Schematic overview of the two-step approach used to genome-edit TERT promoter mutations in hESCs. First, TERT knock-out cell line (TERT$^{\Delta/\Delta}$) that lacks 1.5 kb upstream and 66 bp downstream of the first ATG was established using two CAS9/sgRNAs (sg-1 and sg-15). Second, an sgRNA against the newly synthesized NHEJ-derived junction (−1462 and +67: sg1+15; see *Figure 1—figure supplement 1B*) were co-electroporated with donor plasmids containing the deleted regions with or without the cancer-associated TERT promoter mutations. (**B**) Sequence analysis of targeted cells confirmed successful restoration and introduction of the TERT promoter mutations. (**C**) Telomeric repeat amplification protocol (TRAP) assay of whole cell extracts from TERT$^{\Delta/\Delta}$ hESC lines (n = 2) using 200 ng protein. TERT$^{\Delta/\Delta}$ #1 and #2 cells were collected at day 89 and day 146 after the first editing respectively. IC: internal control. (**D**) Telomere restriction fragment assay of wild-type (WT), TERT$^{\Delta/\Delta}$, and the targeted hESCs over a time course after targeting (day 0: first editing step, day 73: second editing). TERT$^{\Delta/\Delta}$ #1 cells are telomerase-deficient, undergo telomere shortening and die around day 120 unless they regain telomerase activity through the second targeting step. At the first time point (day 101), the majority of the cells are untargeted TERT$^{\Delta/\Delta}$ cells, therefore telomere length is heterogeneous and short. This short telomere length results in reduced hybridization intensity with the TTAGGG radioactive probe. In contrast at the second time point (day 129), uncomplemented TERT$^{\Delta/\Delta}$ #1 died due to progressive telomere shortening and the targeted populations are enriched. In this targeted population the restoration of telomerase resulted in substantial telomere elongation and an

*Figure 1. continued on next page*

*Figure 1. Continued*

overall increase in telomere signal intensity. 2 µg of genomic DNA after digestion with MboI and AluI were loaded in each lane. Quantification of the average telomere length signal is indicated at the bottom of the gel. Throughout all figures we refer to non-targeted wild-type WIBR#3 hESCs as WT. We refer to wild-type cells generated by reintroducing the wild-type promoter into TERT$^{\Delta/\Delta}$ as wt. (**E**) Relative expression levels of TERT mRNA by mutant and wt promoter-containing hESCs over a time course after targeting measured by quantitative RT-PCR. Expression is relative to WT hESCs (black line). Expression of TERT was normalized to GAPDH. Also shown is TERT$^{\Delta/\Delta}$ cells (green line) until day 123. This is the last time point in which RNA could be isolated before TERT$^{\Delta/\Delta}$ cultures died. (**F**) TRAP assay of whole cell extracts from WT and promoter mutation-containing hESCs (day 147) using decreasing amount of protein (200, 40, 8 ng).

The following figure supplements are available for figure 1:

**Figure supplement 1**. Genotyping of TERT$^{\Delta/\Delta}$ hESCs prior to the second targeting that introduced the mutated promoter sequences.

**Figure supplement 2**. Independent confirmation of promoter mutation experiments (shown in *Figure 1C,D*) using an independent TERT$^{\Delta/\Delta\#2}$ cell line.

**Figure supplement 3**. The clonal analysis of TERT promoter mutation containing hESCs confirmed the results of the bulk analysis.

TERT expression when compared to the isogenic wild-type control. This increase is in agreement with previous reports that evaluated these mutations using Luciferase reporter constructs (*Horn et al., 2013*; *Huang et al., 2013*). Noticeably, the two other mutations did not result in similarly increased TERT expression in hESCs. We confirmed this finding for an independent TERT$^{\Delta/\Delta}$ cell line (*Figure 1—figure supplement 2*) and individual single cell-derived hemizygously targeted clones (*Figure 1—figure supplement 3*). Interestingly, the promoter mutation cell lines carrying the −124C/T mutation had elevated levels of TERT mRNA expression, without a equivalent increase in telomerase activity (*Figure 1F*).

## TR, but not TERT is limiting for telomerase in hESCs

A lack of a significant change in telomerase activity despite increased TERT levels in hESCs that carry the −124C/T mutations suggested that TERT mRNA levels are not rate-limiting for telomerase activity in hESCs. Similar observations were made previously for some tumor cell lines in which telomerase activity is limited by levels of TR (*Cristofari and Lingner, 2006*; *Xu and Blackburn, 2007*), and might explain the tissue-specific impact of TERT and TR mutations in patients with dyskeratosis congenita (*Batista et al., 2011*; *Strong et al., 2011*; *Armanios and Blackburn, 2012*). Telomerase biogenesis is a complex biological process that has been shown in human pluripotent stem cells to depend on several activities that, when depleted, can become limiting (*Yang et al., 2008*; *Batista et al., 2011*; *Batista and Artandi, 2013*). To test the hypothesis that in wild type human pluripotent stem cells TR is the limiting factor for telomerase activity, we ectopically expressed TERT, TR, or both from the AAVS1 safe harbor locus (*Hockemeyer et al., 2009*, *2011*) (*Figure 2A*). The introduction of such transgenes into this locus in isogenic settings overcomes concerns of random integration of the transgene. Overexpression levels were verified by western and northern blotting and qRT-PCR (*Figure 2B,C*; *Figure 2—figure supplement 1A*) and quantitative analysis showed that TERT mRNA was overexpressed >40 fold and TR levels by approximately 20 fold. TERT protein was detectable by immunoblotting when overexpressed, contrasted to the lack of detectable endogenous TERT protein. In addition, we determined telomerase activity levels (*Figure 2D* and *Figure 2—figure supplement 1B,C*) and telomere length changes in hESCs 36 days after targeting (*Figure 2E*). TR overexpression strongly increased telomerase activity and led to rapid telomere elongation in hESCs, while overexpression of TERT alone did neither. However, when differentiated into fibroblasts or neural precursor cells (NPCs), we observed the inverse behavior. In this setting, telomerase activity levels were significantly increased when TERT was overexpressed while increased levels of TR did not affect telomerase activity (*Figure 2D*). This finding showed that in hESCs TERT levels were not limiting, and that increased TERT expression did not result in a significant increase of telomerase activity or telomere length. Hence, hESCs are unlikely to reveal the impact of the TERT

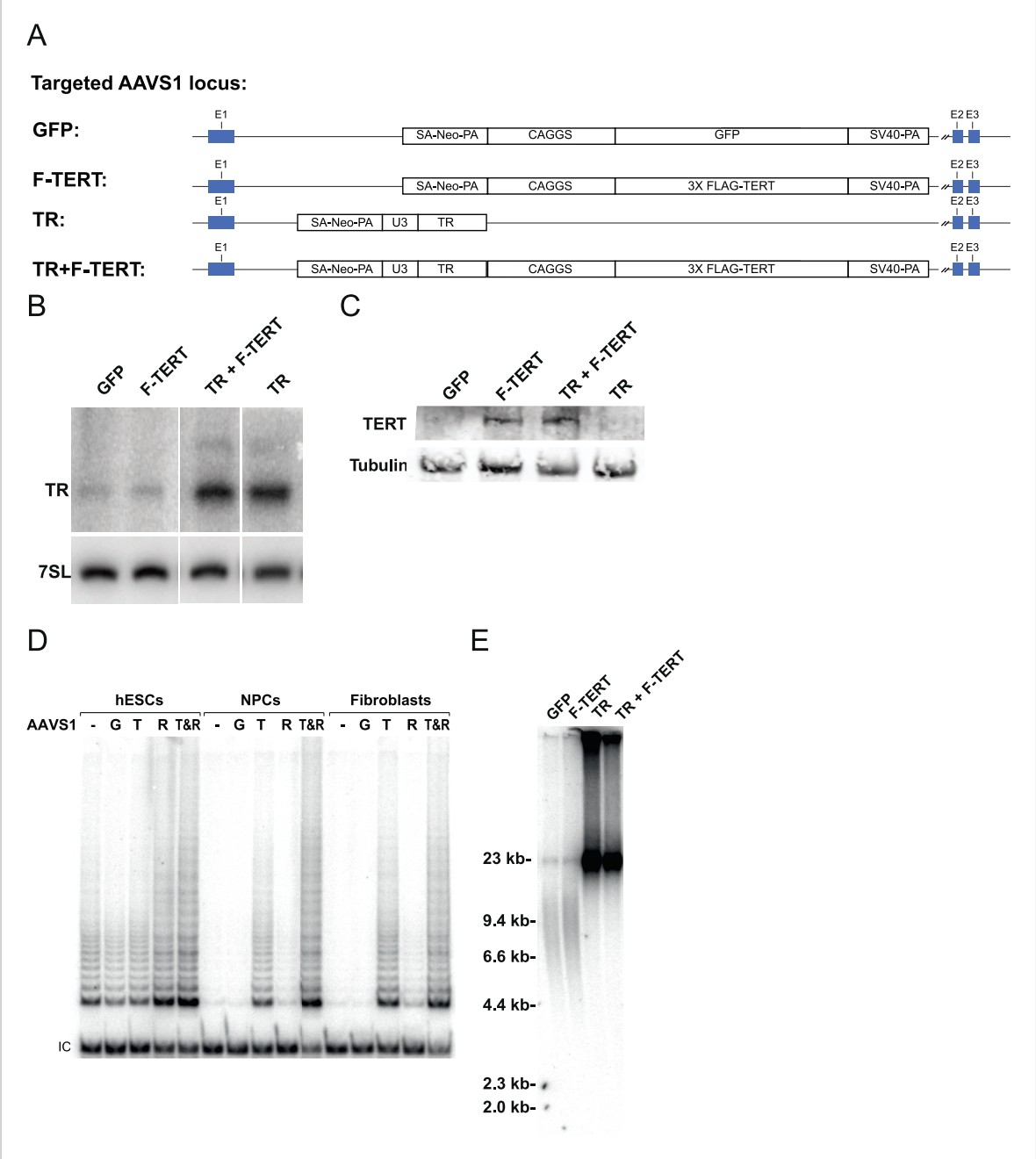

**Figure 2**. Telomerase activity is restricted by levels of TERT in differentiated cells while TR is limiting in wild-type hESCs. (**A**) Targeting schematic of GFP, 3XFLAG-TERT (F-TERT), TR, and F-TERT+TR overexpression from the AAVS1 locus in wild-type hESCs. (**B**) Northern blot detection of total TR and 7SL in targeted hESC lines. TR runs as a doublet in UREA PAGE. (**C**) SDS-PAGE immunoblot of total TERT and tubulin proteins in editing hESC lines from whole cell extract. (**D**) TRAP assay of whole cell extracts from NPCs and fibroblast-like cells differentiated from GFP (G), F-TERT (T), TR (R), or F-TERT+TR (T&R) overexpressing hESCs using 200 ng protein. (**E**) Telomere restriction fragment assay of GFP, F-TERT, TR, and F-TERT+TR overexpressing hESCs.

The following figure supplement is available for figure 2:

**Figure supplement 1**. Quantification of TERT and TR expression levels and telomerase activity in the overexpression hESCs.

promoter mutations. Therefore, observation of the effect of TERT promoter mutations requires the analysis of differentiated cells in which TERT down-regulation results in it becoming limiting for telomerase activity.

## TERT promoter mutations abrogate TERT silencing and impair telomere shortening

We differentiated edited hESCs into embryonic bodies (EBs) and eventually into fibroblasts and determined TERT mRNA levels over a 15 day differentiation period (*Figure 3A*). All cell lines differentiated with equal efficiencies, as evidenced by up-regulation of the differentiation marker COL1A1 and repression of OCT4 transcription (*Figure 3B,C*). Although TERT expression was successfully down-regulated in cells with wild-type TERT promoter, all three promoter mutation lines retained significant levels of TERT expression (*Figure 3A*). This failure of TERT transcriptional silencing became apparent as early as 3 days after the induction of differentiation and accumulated into a fourfold increase in TERT expression in cells that carried the −57A/C or the −146C/T mutation and an 8–12-fold increase in cells in which transcription depended on the endogenous TERT promoter with the −124C/T mutation (*Figure 3A,B*). This failure to appropriately repress TERT transcription during EB differentiation became even more apparent when the cells were differentiated into fibroblast-like cells. As expected, TERT transcription was undetectable in differentiated wild-type fibroblasts. In contrast, fibroblasts with the cancer-associated promoter mutations showed high levels of TERT expression (*Figure 3D*). This difference was not due to impaired differentiation of cells with the TERT promoter mutations, as these cells had silenced OCT4 and appropriately induced COL1A1 expression (*Figure 3D*). Importantly, while telomerase activity is not detectable in wild-type fibroblasts, the aberrant TERT expression resulted in robust telomerase activity in fibroblasts that contained the promoter mutations (*Figure 3E*). As before, we confirmed these findings in an independent TERT$^{\Delta/\Delta}$ cell line (*Figures 3—figure supplement 1A–D*) and in individual single cell-derived targeted clones (*Figures 3—figure supplement 1E,F*). Furthermore we confirmed that this failure to repress TERT expression persists in fibroblasts as late as 45 days after differentiation (*Figures 3—figure supplement 2*).

The failure to silence telomerase could be specific to the fibroblast differentiation paradigm or a more general defect during differentiation. To address this issue, we first generated NPCs using the highly robust dual SMAD inhibition protocol (*Chambers et al., 2009*), establishing NPCs that can be maintained in culture for extended periods of time with low levels of telomerase expression. These NPCs can be further differentiated towards terminally differentiated non-proliferating post mitotic neurons that are characterized by the expression of the pan-neural marker proteins TUJ1 and NEUN. Independent of their genotype, all cells were able to differentiate into NPCs and neurons showing equal down-regulation of OCT4 expression and induction of neuronal marker genes (*Figure 4A–C*). However, a striking difference became apparent in TERT levels as both NPCs and neurons that carried the TERT promoter mutations failed to repress TERT transcription (*Figure 4A,C* and *Figures 4—figure supplement 1*) and showed robust telomerase activity (*Figure 4D*). Even when neurons were maintained in the presence of a mitotic inhibitor, the promoter mutations led to elevated TERT mRNA and telomerase activity levels, suggesting that telomerase activity can accumulate in slowly and non-dividing cells as late as 1 month after induction of terminal differentiation (*Figure 4C,D*).

## Cancer-associated TERT promoter mutations result in telomerase levels equal to those found in immortal tumor cell lines

Next, we assessed the impact of the aberrant telomerase activity of TERT promoter-mutation-containing cells by directly comparing the telomerase levels in fibroblasts and NPCs that carried the promoter mutations to the telomerase activity found in three established tumor cell lines (*Figure 5A,B*). Remarkably, telomerase activity levels in −124C/T NPCs were equivalent to the activity found in immortal Hela S3 cells and about 50% of the activity found in hESCs, HCT116, and 293T cells. Importantly, telomerase activity in these cells was greatly increased relative to wild type cells. Moreover, −124C/T fibroblasts had about 50% of the telomerase activity measured in HeLa S3 cells, 30% of that found in 293T cells, and 25% of that of HCT116 colon carcinoma cells. These findings suggested that TERT promoter mutations induce telomerase levels that are sufficient to enable immortalization or at least significantly delay telomere length-induced senescence. Finally, we analyzed the functional consequences of increased TERT expression by evaluating telomere length changes in hESCs, NPCs, and fibroblasts derived from the isogenic set of TERT promoter-edited cell lines (*Figure 5C,D*, and *Figure 5—figure supplement 1A*). All differentiated cell lines with cancer-associated promoter mutations showed an increase in telomere length compared to the wild-type controls.

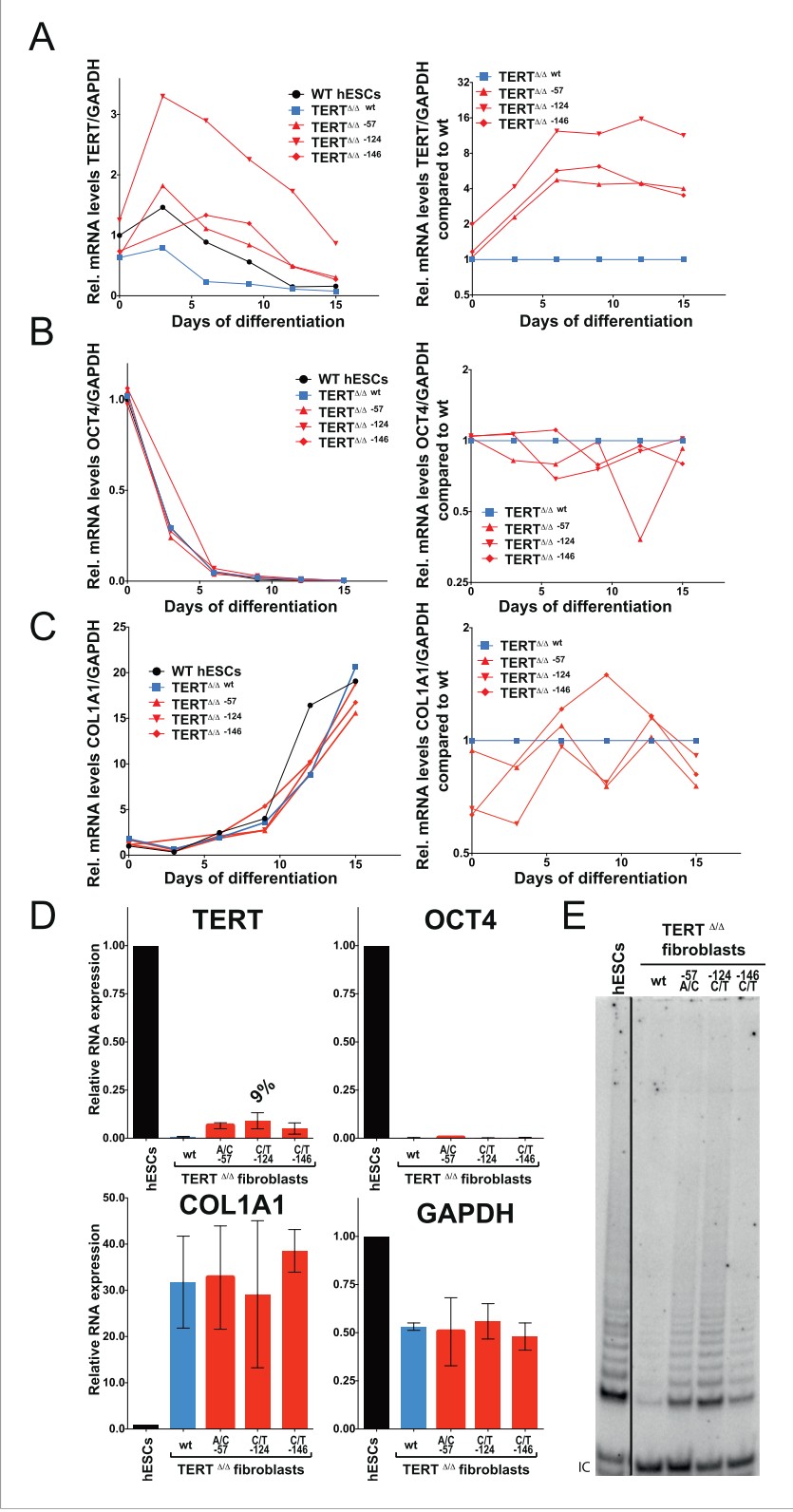

**Figure 3**. Fibroblasts carrying cancer-associated TERT promoter point mutations failed to silence TERT expression upon differentiation and have telomerase activity. (**A**), (**B**) and (**C**) Relative expression level of TERT, OCT4 or COL1A1 in the promoter-mutated hESC-derived fibroblasts compared to WT hESCs over a time course of differentiation (left panel). Relative expression level of TERT, OCT4 or COL1A1 compared to TERT$^{\Delta/\Delta}$ and WT

*Figure 3. continued on next page*

*Figure 3. Continued*

fibroblasts over a time course of differentiation. The right panel shows the same data as in the left panels, normalized to TERT$^{\Delta/\Delta}$ $^{wt}$ fibroblasts. Expression of TERT, OCT4, or COL1A1 was normalized to GAPDH. (**D**) Quantitative RT-PCR of TERT, OCT4, COL1A1, and GAPDH in the fibroblasts carrying the promoter mutations 24 days after differentiation. Expression level is relative to WT hESCs. (**E**) TRAP assay of whole-cell extracts from WT hESCs, and the fibroblasts carrying the TERT promoter mutations (24 days after differentiation) using 2 μg of protein.

The following figure supplements are available for figure 3:

**Figure supplement 1**. Independent confirmation of the failure of TERT repression and telomerase activity upon fibroblast differentiation shown in *Figure 3* using an independent TERT$^{\Delta/\Delta\#2}$ cell line.

**Figure supplement 2**. The failure of TERT repression in fibroblasts was retained throughout long-term culture.

## TERT promoter mutations suppress telomere shortening in tumors

To further explore the in vivo relevance of these findings in the context of long-term differentiation as well as tumor progression, cells with TERT promoter mutations were assayed for teratoma tumor formation in immune-compromised mice (*Figure 5E* and *Figure 5—figure supplement 1B*). For this assay all cell lines were injected subcutaneously into NOD/SCID mice, allowing pluripotent cells to differentiate and form a teratoma comprised of cells derived from all three germ layers. All cell lines injected formed teratomas of approximately equal size, were explanted simultaneously (75 days after injection), and analyzed for their telomere length. Using this unbiased approach, we again found that cells with TERT promoter mutations carried aberrantly long telomeres, with the −124C/T mutation having the strongest defect in silencing telomerase activity and retaining almost identical telomere length as undifferentiated hESCs (*Figure 5C,E* and *Figure 5—figure supplement 1B*). These data demonstrated the causal relationship between the TERT promoter mutations and telomere maintenance and showed that the TERT promoter mutations can up-regulate TERT levels sufficiently to suppress telomere erosion without additional tumor-selected changes.

## Discussion

### Cancer-associated mutations affect TERT upon differentiation

A key challenge in cancer research is to understand how mutations that sequentially occur in normal cells eventually produce a tumor. For noncoding mutations identified by GWAS, this is a particular challenge. Here we were able to dissect how the most frequent noncoding mutations in human cancer exert their tumorigenic effect. We are able to do so because cancer genomics have identified candidate mutations, genome editing is facile and robust, and we tested the effects in an otherwise wild-type background, thus being able to attribute a phenotypic effect specifically to a single genetic change.

Using genome editing of the endogenous TERT locus we generated a panel of three hESC lines that differed exclusively at a single position in the TERT promoter associated with cancer. Analyzing the impact of these mutations in hESCs, we showed that the most frequent mutation, −124C/T, increased TERT mRNA levels in hESCs by about 2–3 fold. However, neither the −57A/C nor the −146C/T mutation led to an increase in TERT transcription, despite the fact that these mutations generate the same putative ETS-binding motif of TTCCGG. This suggests a strong positional effect between the location of the ETS mutation and the core transcription initiation machinery. The possibility of such context-dependent positional constraints between the TERT promoter mutations and the core transcriptional machinery is further supported by the fact that a single point mutation at position −89 (C/G, −31 bp upstream of the TSS) would result in the same TTCCGG sequence. The fact that this mutation has as of now not been reported to be associated with cancer despite it being between the −57 and −124 sites is likely due to the need of the core transcription factors to bind to this site. It is intriguing to speculate why all promoter mutations are located in close proximity to the TSS; it seems possible that TERT transcriptional regulation is closely linked with the core

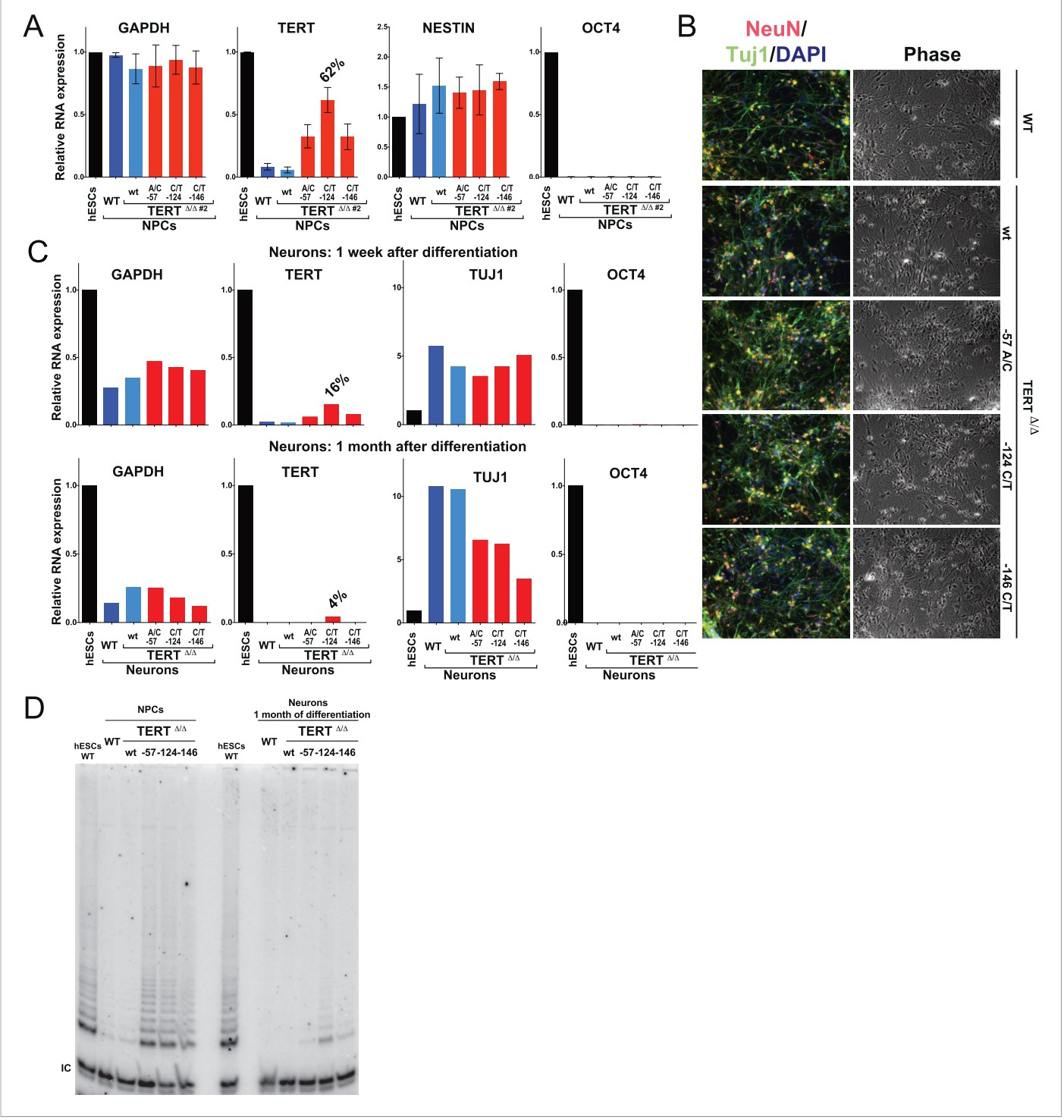

**Figure 4**. Neural precursors and neurons differentiated from the promoter mutation hESCs failed to repress TERT and telomerase activity. (**A**) Quantitative RT-PCR of GAPDH, TERT, NESTIN, and OCT4 in the neural precursors carrying the promoter mutations 20–25 days after differentiation from hESCs. Expression levels are relative to the WT hESCs. (**B**) Phase-contrast and immunofluorescence images of neurons differentiated from wild-type hESCs or the TERT promoter mutation-containing hESCs. Shown are cells 28 days after neural induction from NPCs and treated with mitotic inhibitor for 16 days. The left panel shows IF staining against NeuN (red), Tuj1 (green), and DAPI staining (blue). (**C**) Quantitative RT-PCR of GAPDH, TERT, TUJ1, and OCT4 in the neurons carrying the promoter mutations. The top panel shows expression levels of neurons 7 days after neuronal differentiation from NPCs. The bottom panel shows expression levels of neurons 28 days after induction of neuronal differentiation from NPCs and treated with mitotic inhibitor

*Figure 4. Continued*

for 16 days. Expression level is relative to the WT hESCs. (**D**) TRAP assay of whole cell extracts from NPCs (35 days after differentiation from hESCs) and neurons (28 days after neuronal differentiation from NPCs and treated with mitotic inhibitor for 16 days) using 1 µg protein.

The following figure supplement is available for figure 4:

**Figure supplement 1**. Clonal analysis of TERT promoter mutation NPCs confirmed results from bulk analysis.

transcriptional machinery rather than regulated through the canonical positioning of transcription factors along an extended promoter. The experimental approach established in this study of genome editing the TERT promoter provides an experimental system to uncover *cis*-regulatory elements that are necessary for telomerase expression in stem cells and its transcriptional regulation upon differentiation.

Increased expression of TERT mRNA in −124C/T containing hESCs did not lead to a significant increase in telomerase activity or pronounced telomere lengthening, establishing that in hESCs TR levels, but not TERT levels, are limiting for telomerase assembly and telomere lengthening. Therefore, immortal hESCs are as uninformative with regard to cancer-associated TERT mutations as immortal tumor tissue or cell lines. However, whereas upon differentiation wild-type hESCs efficiently silence TERT transcription, resulting in loss of telomerase activity and telomere shortening, the cancer-associated TERT promoter mutations were sufficient to maintain expression of TERT and resulted in telomerase activity levels comparable to immortal cancer cell lines. These experiments uncover that the underlying cancer-causing mechanism is likely a failure to repress telomerase upon differentiation into somatic cells. It is remarkable that TERT promoter mutations are sufficient to up-regulate TERT expression without additional cancer-selected changes in the genome such as increased levels of ETS factors.

## An explanation for the tumor spectrum of TERT promoter mutations

TERT promoter mutations are not frequently found in leukemias and colorectal cancers (*Heidenreich et al., 2014*). Direct evidence (*Chiu et al., 1996*; *Schepers et al., 2011*) as well as the pathology of the telomerase-related disease dyskeratosis congenita in which patients with mutations in the telomere maintenance pathway present with bone marrow failure as well as lung, intestinal, and skin pathologies show that TERT is expressed in these highly proliferative tissues and is required for their long-term self-renewal capacity and ability to maintain tissue homeostasis (*Armanios and Price, 2012*; *Aubert et al., 2012*).

Tumor-initiating events in these cancers predominantly drive proliferation pathways that spur formation of hyperplasia and niche-independent proliferation that allow incipient cancer cells to outcompete their neighbors (*Barker et al., 2009*; *Zhou et al., 2009*; *Merlos-Suarez et al., 2011*; *Magee et al., 2012*). In this setting, mutations in the TERT promoter or alterations in the telomerase biogenesis pathway might be at first neutral, not providing a direct proliferative advantage as telomeres are still long or telomerase is active (*Figure 6A*). During the genesis of these tumors, telomere shortening might present a challenge at a later stage when cells have already outcompeted their neighbors.

In contrast, tumor-initiating cells that are thought to not directly arise from a canonical telomerase-positive stem cell compartment (e.g., liposarcomas), that undergo high numbers of divisions after differentiating (e.g., neural-crest derived melanocytes), or that have to reenter a proliferation cycle in response to chronic injury (e.g., urothelial cells and hepatocytes) could be challenged by the telomere-dependent proliferative barrier comparatively early in their progression. In these cell types TERT promoter mutations will provide an immediate and strong proliferative advantage over neighboring cells. In this case telomerase activation occurs in cells in which telomerase is absent or low and which have an otherwise mostly intact genome. In these cells activating TERT promoter mutations will be present in most tumor cells and detected as a frequent and thus early event (*Figure 6B*).

It is important to note that this model depicts the very extreme cases of a TERT-positive adult stem cell with long telomeres contrasted to the expected outcome of a TERT promoter mutation in a telomerase-negative cell with short telomeres. Likely this sharp distinction between canonical telomerase-positive stem cell compartments and telomerase-negative compartments is rather

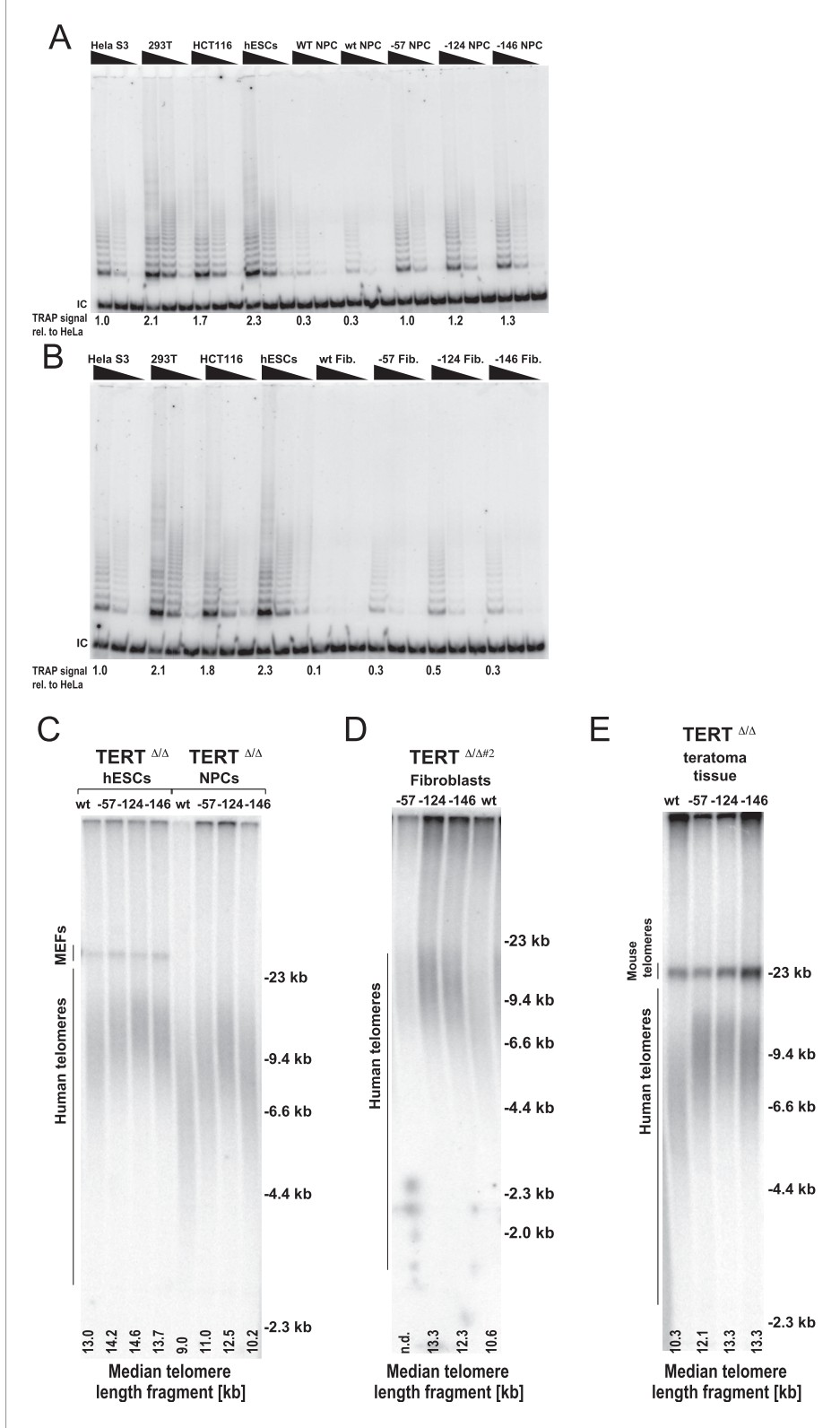

**Figure 5**. Fibroblasts and neural precursors carrying cancer-associated TERT promoter point mutations showed comparable telomerase activity to cancer cell lines, and telomere length was maintained over long-term culture and tumor development. (**A**) and (**B**) TRAP assay of whole-cell extracts from cancer cell lines (HeLaS3, 293T, and HCT116), WT hESCs, and the NPCs or fibroblasts carrying the TERT promoter mutations (24 days after differentiation for
*Figure 5. continued on next page*

*Figure 5. Continued*

fibroblasts and 20 days for NPCs) using decreasing amount of protein (200, 40, 8 ng). For comparison, TRAP samples in (**A**) and (**B**) were prepared simultaneously and samples from cancer cell lines are identical in (**A**) and (**B**). TRAP signals relative to HeLa S3 were quantified and are shown at the bottom of the gels. (**C**) Telomere restriction fragment assay of the hESCs and NPCs (65 days after differentiation from hESCs). Median telomere length signals were quantified and shown at the bottom. It is important to note the telomere shortening in NPCs and fibroblasts in wild type cells exceeds the initial telomere length difference found in the hESCs. (**D**) Telomere restriction fragment assay of the fibroblasts (30 days after differentiation from hESCs). Median telomere length signals were quantified and shown at the bottom. (**E**) Telomere restriction fragment assay of teratoma tumor tissue generated from wt and promoter-mutation containing hESCs (75 days after injection). Median telomere length signals were quantified and shown at the bottom.

The following figure supplement is available for figure 5:

**Figure supplement 1**. Fibroblasts and teratoma tissue carrying cancer-associated TERT promoter point mutations maintained telomere length over long-term culture and tumor development.

---

continuous in vivo. Telomerase expression, telomere length, and the number of cell divisions will differ between tissues and with age and therefore the benefit of the TERT promoter mutation will be complexly graded. Given this, it will be critical to determine exactly which cells of the human body are telomerase-positive, when and how telomerase is silenced upon differentiation, and how many divisions cells undergo in human tissue after becoming telomerase-negative.

## Telomerase inhibition as a cancer treatment

Telomerase inhibition has been proposed as a target for cancer therapies. We demonstrate that TERT promoter mutations are sufficient to de-repress TERT, providing a potential target to inhibit TERT expression and telomerase activity.

In order to identify therapeutic approaches specific to these promoter mutations, a model system in which TERT is dysregulated solely by these mutations is necessary. Our model system fulfills this requirement and allows for a direct assessment of any potential inhibition by measuring TERT expression following differentiation. In contrast, this approach will be challenging in cancer cells, as TERT mRNA levels, telomerase levels, and telomere length vary dramatically regardless of whether they carry any of the TERT promoter mutations. Further mechanistic studies in such tumor cells are also challenged by the high frequency of concurrent TERT copy number variations, promoter polymorphisms, and cancer-associated dysregulation of factors implicated in TERT regulation such as MYC. As such, it will be challenging to evaluate the effectiveness of such an inhibitor due to these potentially compensatory effects arising from these misregulations. As such, it is imperative to test any potential therapeutic approach directed at these promoter mutations in a model system that only carries these mutations in an otherwise wild-type background, such as the model system described here. Specifically targeting the TERT promoter mutations is an attractive approach, as TERT promoter mutations are exclusive to the tumor cells and are not present in surrounding normal tissue. Therefore, any intervention that is targeted specifically against their mode of operation is expected to affect tumor cell survival, but not the telomerase-positive adult stem cells of the patient.

## Material and methods

### hESC culture

Genome-editing experiments were performed in WIBR#3 hESCs (*Lengner et al., 2010*), NIH stem cell registry # 0079. Cell culture was carried out as described previously (*Soldner et al., 2009*). Briefly, all hESC lines were maintained on a layer of inactivated mouse embryonic fibroblasts (MEFs) in hESC medium (DMEM/F12 [Lifetech]) supplemented with 15% fetal bovine serum [Lifetech], 5% KnockOutTM Serum Replacement [Lifetech], 1 mM glutamine [Lifetech], 1% non-essential amino acids [Lifetech], 0.1 mM β-mercaptoethanol [Sigma], 1000 U/ml penicillin/streptomycin [Lifetech], and 4 ng/ml FGF2 [Lifetech]. Cultures were passaged every 5–7 days either manually or enzymatically with collagenase type IV [Lifetech] (1.5 mg/ml) and gravitational sedimentation by washing 3 times in wash

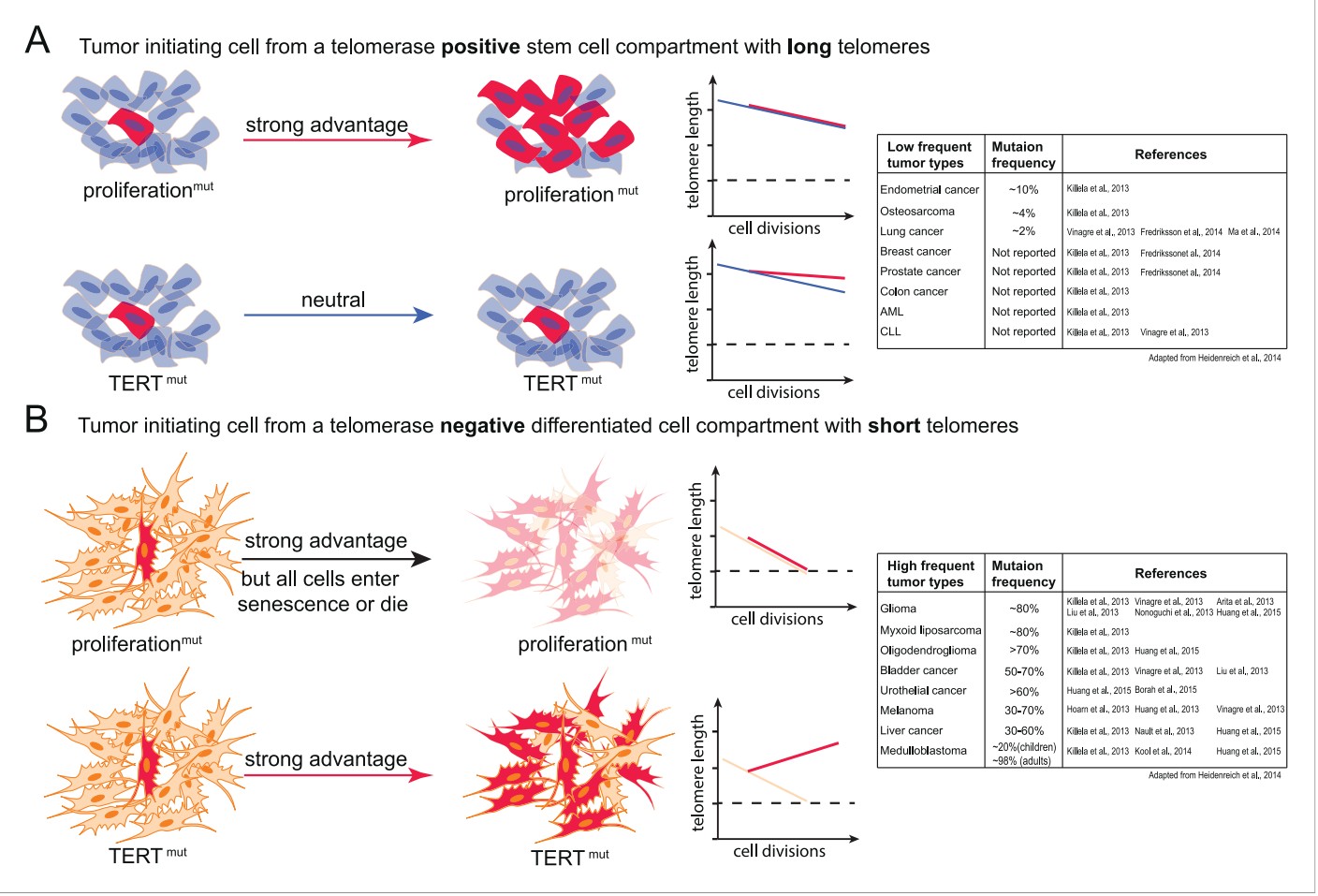

**Figure 6**. Model explaining the tumor spectrum associated with TERT promoter mutations. Shown are the differential outcomes of a cell acquiring a cancer-associated TERT promoter mutation or a proliferation-inducing mutation dependent on telomere length of the cell. (**A**) In a cell with long telomeres and telomerase activity, a proliferation-promoting mutation will result in a strong proliferative advantage and can act as the tumor-initiating event. Cells with long telomeres arise from tissues that have a telomerase positive stem cell compartment such as the hematopoietic or intestinal system. In contrast, mutations in the TERT promoter do not provide a proliferative advantage, they are neutral and do not promote tumor formation. Cell states are depicted on the left; cells that acquire mutations are shown in red. A schematic depicting telomere length changes as a function of the number of cell divisions is shown on the right. The dashed line indicates the critical telomere length at which cells are subjected to the Hayflick limit and stop proliferating or die. The red line indicates the telomere length changes predicted for cells that acquire either a proliferation-promoting mutation (top) or a TERT promoter mutation (bottom). The blue line indicates the telomere length changes predicted for wild-type cells. Indicated is a case where telomere shortening is suppressed by the TERT promoter mutations. However, since these cell already have long telomeres and/or naturally express telomerase, telomeres in neither wild-type cells or cells acquiring a proliferation inducing mutation will shorten to the point that the cells are subjected to the Hayflick limit. (**B**) Schematic as shown in (**A**) but for a telomerase negative cells with short telomeres. A proliferation-promoting mutation will also provide a growth advantage in telomerase-negative differentiated cells with short telomeres, however, these cells will enter replicative senescence or die. In contrast, a cell with short telomeres acquiring a TERT promoter mutation can bypass the Hayflick limit (dashed lines), immortalize, and outcompete its neighboring cells. Cell states are depicted on the left; cells that acquire mutations are shown in red. Schematic depicting telomere length changes as a function of the number of cell divisions is shown on the right. The orange line indicates the telomere length changes predicted for wild-type cells. The table to the right shows the frequency of TERT promoter mutations found in different types of tumors (adapted form *Heidenreich et al., 2014*). The table includes references that report the specific tumor subtypes and frequencies used to generate this table.

media (DMEM/F12 [Lifetech] supplemented with 5% fetal bovine serum [Lifetech], and 1000 U/ml penicillin/streptomycin [Lifetech]).

## Differentiation to fibroblast-like cells

For the formation of EBs hESC colonies were grown on petri dishes in fibroblast medium (DMEM/F12 [Lifetech]) supplemented with 15% fetal bovine serum [Lifetech], 1 mM glutamine [Lifetech], 1%

non-essential amino acids [Lifetech], and penicillin/streptomycin [Lifetech, Carlsbad, CA]. After 9 days EBs were transferred to tissue culture dishes to attach. Fibroblast-like cells were passaged with Trypsin EDTA ([Lifetech], 0.25%), triturated into a single-cell suspension and plated on tissue culture dishes. Cultures were maintained in fibroblast media and passed every 6 days.

## Differentiation to NPCs and neurons

Before differentiation to NPCs, hESCs were cultured under feeder-free conditions on matrigel [Corning]-coated plates in E8 media (DMEM/F12 [Lifetech]) supplemented with 64 µg/ml L-ascorbic acid [Sigma], 19.4 µg/ml insulin [Sigma, St. Louis, MO], 14 µg/l sodium selenite [Sigma], 543 ng/l sodium bicarbonate [Sigma], 1000 U/ml penicillin/streptomycin [Lifetech], 100 ng/ml FGF2 [Lifetech], and 10.7 µg/ml Transferrin [Sigma]. hESCs were passaged with accutase [Invitrogen] and triturated to a single-cell solution and plated on matrigel-coated plates at 50,000 cell/cm$^2$. The dual SMAD inhibition protocol for the differentiation of hESCs to NPCs was adapted from *Chambers et al. (2009)*. Differentiation was induced when cells reached 90–100% confluency.

NPCs were maintained in N2 media (50% DMEM/F12 [Lifetech], 50% Neurobasal Media [Lifetech] supplemented with 0.75% BSA (wt/vol) [Sigma], N2 Supplement [Lifetech], 20 ng/ml insulin [Sigma], 1 mM glutamine [Lifetech], 1000 U/ml penicillin/streptomycin [Lifetech], 25 ng/ml FGF2 [Lifetech] and 40 ng/ml EGF [R&D systems]) and passaged every 5 days. For the terminal differentiation to neurons NPCs were plated at 50,000 cells/cm$^2$ on matrigel-coated plates in N2B27 media (50% DMEM/F12 [Lifetech], 50% Neurobasal Media [Lifetech] supplemented with 0.75% BSA (wt/vol) [Sigma], N2 Supplement [Lifetech], B27 Supplement [Lifetech], 1 mM glutamine [Lifetech], 1000 U/ml penicillin/streptomycin [Lifetech]). Neurons were treated with 250 nM mitotic inhibitor (Cytosine-β-D-arabinofuranoside [Sigma]).

## Gene targeting in hESCs

All targeting experiments were preformed as previously described (*Hockemeyer et al., 2009*, *2011*). CAS9 and all sgRNAs were expressed using the px330 plasmid (*Cong et al., 2013*). Cancer associated TERT promoter mutation containing cell lines were generated by two targeting steps. First, $1–2 \times 10^7$ hESCs were co-electroporated with 15 µg of two CAS9 plasmids targeting −1418 to −1399 bp (aaccgcccctttgccctag) and +110 to +129 bp (taccgcgaggtgctgccgc) from the TSS and 7.5 µg of a GFP-expression plasmid was electroporated along with px330. Cells were sorted for GFP fluorescence 72 hr after electroporation. Single-cell derived hES colonies were isolated and their targeting was confirmed by Southern blotting and PCR followed by sequencing. 120 clones were analyzed and three homozygous targeted hESC lines (TERT$^{\Delta/\Delta}$) were obtained. For the second targeting, px330 plasmids were designed with sgRNAs against the newly formed NHEJ-derived junction site in TERT$^{\Delta/\Delta}$ cells and electroporated with 35 µg of a repair plasmid that carried either the wild type TERT promoter element (wt) or the respective TERT promoter mutations (57A/C, 124 C/T, 146C/T). After the second targeting, cells were continuously passaged. Over a period of 120 days all TERT$^{\Delta/\Delta}$ lines that did not undergo the second targeting step died due to critically short telomeres. However, cells that were correctly targeted in the second targeting step regained TERT expression and outgrew untargeted cells. These cells were analyzed in bulk or as single cell derived clones, after the parental TERT$^{\Delta/\Delta}$ control culture, that did not undergo the second targeting step, had completely died. Targeting of individual clones was confirmed by Southern blot analysis.

## qRT-PCR

RNA was extracted with TRIzol (Lifetech) and treated with DNaseI (NEB). 600 ng RNA were converted to cDNA with the iScript Reverse Transcriptase (BioRad) and random and poly A priming. TR cDNA was prepared by gene specific reverse transcription. qRT-PCR was performed with KAPA SYBR fast [KAPA Biosystems] or SYBR Select Master Mix (ABI) in 96-well or 384-well format with a total reaction volume of 20 µl or 10 µl respectively. 2 µl cDNA from the iScript reaction mixture was used for the detection of TERT mRNA. For measuring the expression levels of all other genes, cDNA was diluted 1:10 and 2 µl were used for qPCR.

Due to different expression levels of GAPDH between hESCs and differentiated cells, GAPDH data are shown in the figures that required comparison of expression in different cell types. Relative expression levels were calculated based on Δ/Δ Ct and/or ΔCt analysis. qRT-PCR primers used in this study are summarized in *Supplementary file 1*.

## Immunofluorescence

For analysis by IF, cells were briefly rinsed with PBS, and fixed with 4% formaldehyde in PBS. Cells were blocked with PBS 0.3% Triton X-100 with 5% horse serum. Fixed cells were incubated with antibodies against NEUN (mouse, monoclonal, [Millipore], MAB377; 1:1500) and TUJ1 (B-III-Tubulin, chicken, polyclonal, [Millipore], AB9354, 1:500), in PBS 0.3% Triton X-100 with 1% BSA over night. After washing with PBS the cells were stained with secondary antibodies (Alexa Fluor 546 goat α mouse, Alexa Fluor 488 goat α chicken [Lifetech]; 1:500), for 1 hr in PBS 0.3% Triton X-100 with 1% BSA. Cells were then washed with PBS and stained with 1 ng/μl DAPI (Sigma) in PBS.

## RNA detection, Southern blotting, and assaying telomerase catalytic activity

RNA for northern blot was purified using TRIzol according to the manufacturer's protocol (Lifetech). Northern blot detection of TR was performed as previously described (*Fu and Collins, 2003*). 7SL RNA was detected using $^{32}$P end-labeled probe (TGAACTCAAGGGATCCTCCAG) under similar conditions as TR, except hybridization took place at 37°C. Southern blots analysis was performed as previously described (*Hockemeyer et al., 2009, 2011*) using a 3′- probe for TERT (6280 bp −6846 bp downstream of the TERT first ATG) and probe T1 (amplified from hES genomic DNA with primers Fw: GTGACTCAGGACCCCATACC and Rev: ACAACAGCGGCTGAACAGTC). PCR-based telomeric repeat amplification protocol (TRAP) was performed as previously described using TS (AATCCGTCGA GCAGAGTT) and ACX (GCGCGGCTTACCCTTACCCTTACCCTAACC) for amplification of telomereic repeats and TSNT (AATCCGTCGAGCAGAGTTAAAAGGCCGAGAAGCGAT) and NT (ATCGCTT CTCGGCCTTTT) as an internal control (*Kim et al., 1994*). Real-time quantitative telomeric repeat amplification (QTRAP) was performed similar to previously published protocols (*Wege et al., 2003*). Cell extract was generated from CHAPS lysis and samples were normalized using the BCA Protein Assay Kit (Pierce). 200 ng of total protein was used per 20 μl QTRAP reaction, which was composed of iTaq Universal SYBR Green Supermix (Bio-Rad) and 0.1 μg TS and 0.02 μg ACX primers. Samples were incubated at 30°C for 30 min before a 2 min 95°C hot-start and 35 cycles of 95°C for 15 s and 61°C for 90 s. Relative telomerase activity was calculated by ΔCt to the reference sample.

## Immunoblotting

After heating to 80°C for 5 min, protein samples were cooled to room temperature and resolved by SDS-PAGE. Protein was then transferred to nitrocellulose membrane and subsequently incubated with mouse α-tubulin (1:500, DM1A, [Calbiochem]) and mouse anti-TERT (1:3000 [Geron]) in 4% nonfat milk (Carnation) in TBS buffer (150 mM NaCl, 50 mM Tris pH 7.5) overnight at 4°C. The membrane was washed in TBS and incubated with goat α-mouse Alexa Fluor 680 (1:2,000, [Life Technologies]) in 4% nonfat milk in TBS for 1 hr at room temperature. After extensive washing with TBS, the membrane was visualized on a LI-COR Odyssey imager (*Fu and Collins, 2003*).

## Detection of telomere length

For preparation of genomic DNA, hESC lines were washed with PBS, released from the feeder cell layer by treatment with 1.5 mg/ml collagenase type IV and washed 3× in wash media by gravitational sedimentation to minimize contaminating MEF cells. Genomic DNA was then prepared as described previously (*Hockemeyer et al., 2005*). While this method removes the vast majority of MEFs, the signal from mouse telomeres is disproportionate to human telomeres due to amplified relative length and concentration into a smaller area (*Kipling and Cooke, 1990*). Because MEF telomeres are size-resolved from human telomeres they do not interfere with analysis of hESC telomere length. Genomic DNA was digested with MboI and AluI overnight at 37°C. The resulting DNA was normalized and run on 0.75% agarose (Seakem ME Agarose, Lonza), dried under vacuum for 2 hr at 50°C, denatured in 0.5 M NaOH, 1.5 M NaCl for 30 min, shaking at 25°C, neutralized with 1 M Tris pH 6.0, 2.5 M NaCl shaking at 25°C, 2× for 15 min. Then the gel was pre-hybridized in Church's buffer (1% BSA, 1 mM EDTA, 0.5M NaP0$_4$ pH 7.2, 7% SDS) for 1 hr at 55°C before adding a $^{32}$P-end-labeled (T$_2$AG$_3$)$_3$ telomeric probe. The gel was washed 3× 30 min in 4× SSC at 50°C and 1× 30 min in 4× SSC + 0.1% SDS at 25°C before exposing on a phosphorimager screen.

## Teratoma tumor formation assays

Teratoma formation assays where performed as previously described in (*Hockemeyer et al., 2008*).

## Acknowledgements

We thank Manraj Gill and Kartoosh Heydari for technical assistance and the Collins and Hockemeyer labs for helpful discussion and Kathleen Collins and Dan Youmans for AAVS1 expression constructs. Kunitoshi Chiba is supported by a Nakajima foundation fellowship. Dirk Hockemeyer is a New Scholar in Aging of the Ellison Medical Foundation and is supported by the Glenn Foundation as well as the The Shurl and Kay Curci Foundations.

## Additional information

### Funding

| Funder | Grant reference | Author |
| --- | --- | --- |
| Ellison Medical Foundation | | Dirk Hockemeyer |
| Glenn Foundation for Medical Research | | Dirk Hockemeyer |
| National Cancer Institute (NCI) | R01 RCA196884A | Kunitoshi Chiba, Joshua Z Johnson, Jacob M Vogan, Tina Wagner, Dirk Hockemeyer |

The funders had no role in study design, data collection and interpretation, or the decision to submit the work for publication.

### Author contributions

KC, JZJ, TW, DH, Conception and design, Acquisition of data, Analysis and interpretation of data, Drafting or revising the article; JMV, Conception and design, Acquisition of data, Analysis and interpretation of data; JMB, Performed the teratoma formation assays, Acquisition of data

### Author ORCIDs

John M Boyle, http://orcid.org/0000-0001-6173-6765

### Ethics

Animal experimentation: This study was performed in strict accordance with the recommendations in the Guide for the Care and Use of Laboratory Animals of the National Institutes of Health. All of the animals were handled according to approved institutional animal care and use committee (IACUC) protocols (R355-0114) of the University of California Berkeley.

## Additional files

### Supplementary file

• Supplementary file 1. Summary of qRT-PCR primers used in this study.

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
