## [Decision Letter]

Thank you for submitting your work entitled “Cancer-associated TERT promoter mutations abrogate telomerase silencing” for peer review at *eLife*. Your submission has been favorably evaluated by Sean Morrison (Senior editor) and three reviewers, one of whom served as guest Reviewing editor. One of the peer reviewers, Steven Artandi, has agreed to reveal his identity.

The reviewers have discussed the reviews with one another and the Reviewing editor has drafted this decision to help you prepare a revised submission.

The three reviewers agree that this is an excellent and rigorously performed study providing a compelling and exciting narrative regarding the effects of highly recurrent mutations in the TERT promoter associated with many human cancers.

The manuscript addresses the 20-year old question of how telomerase is activated in human cancer. From the seminal work of Garraway and Kumar a few years ago, it is clear that one route is through mutations in the hTERT promoter that create ETS binding sites. It was proposed that ETS transcription factor augment expression of hTERT and indeed a 2-4 fold increase in expression was noted with the mutant promoter linked to luciferase in comparison with the wild type form of the promoter (transfection experiments in cancer cell lines). From work on dyskeratosis congenita, it is known that a 2-fold reduction in telomerase can have major impact but whether a mild increase could explain the tumorigenic potential of the mutations has not been established.

The current work examines the mechanism by which the promoter mutations act using a two-step CRISPR knockin approach in human ES cells and then studying the effect of the mutations on telomerase and telomere length maintenance upon differentiation. The striking observation is that the mutant promoter becomes resistant to the silencing that normally occurs upon ES cell differentiation. Thus, the promoter mutations act by abrogating the off-switch that normally ensures telomerase silencing, thwarting the telomere shortening program that limits tumorigenesis. These results and conclusions are highly significant to the cancer field.

Essential revisions:

1) The manuscript examines the effect of TERT promoter mutations with respect to telomerase activity and telomere length. Figure 3 shows the hTERT expression is likely to continue to decrease after 15 days if it continues on the same trajectory. Is 15 days the right time point? It remains unclear whether these mutations have the capacity to shift the Hayflick limit or delay/bypass senescence. Do the authors have information on whether these fibroblasts are truly immortal?

2) The presentation of the model attempting to explain why TERT promoter mutations occur in certain tumor types, but not in others, could perhaps be more nuanced, and may be a little premature since additional TERT activating mutations may surface from further analysis in cancer that are currently thought to not fall in this category. Also, the authors draw a sharp distinction between “canonical telomerase-positive stem cell compartments” in which TERT promoter mutations would not be beneficial, and others in which they would confer selective advantage. Given our dearth of knowledge about the lineage hierarchy in many organs, these statements should perhaps be worded more cautiously. Also, it would be helpful the paper should contain a figure (or a figure adaptation) specifically depicting the frequency of TERT mutations across all human cancer types for the edification of the reader. Otherwise, it is difficult to determine to what degree the model in Figure 6 is explains the data.

---

## [Author Response]

*1) The manuscript examines the effect of TERT promoter mutations with respect to telomerase activity and telomere length.*
Figure 3
*shows the hTERT expression is likely to continue to decrease after 15 days if it continues on the same trajectory. Is 15 days the right time point? It remains unclear whether these mutations have the capacity to shift the Hayflick limit or delay/bypass senescence. Do the authors have information on whether these fibroblasts are truly immortal*?

The reviewers raise a very important point. The reviewers point out that hESCs that carry the TERT promoter mutation still down-regulate TERT levels upon differentiation. However, unlike wild-type cells, they fail to fully silence TERT levels in differentiated cells. As TERT levels do not seem limiting in hESCs the reduction of TERT levels is not directly proportional to telomerase activity, so that the residual TERT levels (5-9% compared to hESCs) in TERT promoter mutation-containing cells levels result in levels of telomerase that are equivalent to the levels found in immortal tumor cells. To address the reviewers’ concern that that TERT levels might further decrease and that TERT is fully silenced even in TERT promoter mutation-containing cells, we have now included additional RT-PCR data that analyze TERT levels 40 and 45 days after differentiation. Here, we find that all three TERT promoter mutation-containing cell lines also maintain TERT levels of about 5% compared to hESCs even at the late time points of differentiation and have long telomeres. We fully agree with the reviewers that TERT levels and telomere length maintenance are indirect measures for immortality and that it is not clear how many additional cellular divisions are ultimately granted by these additional TERT promoter mutations. We appreciate the reviewers’ concern that we are not directly testing immortalization and only provide indirect evidence.

Using these points, we have revised the Discussion appropriately.

*2) The presentation of the model attempting to explain why TERT promoter mutations occur in certain tumor types, but not in others, could perhaps be more nuanced, and may be a little premature since additional TERT activating mutations may surface from further analysis in cancer that are currently thought to not fall in this category. Also, the authors draw a sharp distinction between* “*canonical telomerase-positive stem cell compartments*” *in which TERT promoter mutations would not be beneficial, and others in which they would confer selective advantage. Given our dearth of knowledge about the lineage hierarchy in many organs, these statements should perhaps be worded more cautiously. Also, it would be helpful the paper should contain a figure (or a figure adaptation) specifically depicting the frequency of TERT mutations across all human cancer types for the edification of the reader. Otherwise, it is difficult to determine to what degree the model in*
Figure 6
*is explains the data*.

We appreciate the reviewers’ comments and fully agree that it would be better at this point to discuss our model more cautiously. We have revised the text so that it becomes clear that telomerase expression, telomere length, and number of cell divisions are expected to differ between tissues and that therefore the benefit for the TERT promoter mutation will be complexly graded. As the reviewers correctly point out, our model presents the extreme cases of a telomerase-negative differentiated cell with short telomeres compared to a telomerase-positive adult stem cell with long telomeres. Following the reviewers’ suggestion we now include an adaptation of a table that describes the frequencies with which TERT promoter mutations are found in different cancer types.